Intra- and inter-rater reliability of ultrasound measurements of the levator ani muscle thickness in and M-mode in asymptomatic individuals: a cross-sectional study

http://orcid.org/0000-0002-0730-0079 Delgado-Pérez Esther 1
http://orcid.org/0000-0001-9949-5178 Bermejo-Franco Alberto 1 2
Díaz-Meco Raquel 1
http://orcid.org/0000-0002-1782-2524 Abuín-Porras Vanesa 1
http://orcid.org/0000-0001-6598-829X Romero-Morales Carlos 1
González-Fernández Laura 1 laura.gonzalez@universidadeuropea.es
De-La Cueva-Reguera Mónica 1
1 Department of Physiotherapy, Faculty of Medicine, Health and Sports, European University of Madrid , Villaviciosa de Odón, Madrid , Spain
2 Neurosciences & Physical Therapy Research Group. Department of Physiotherapy, Faculty of Medicine, Health and Sports, European University of Madrid , Villaviciosa de Odón, Madrid , Spain
Tomar Mahendra
Electronic publication date: 2025 Jun 24
Publication date: 2025
Volume: 13
Electronic Location ID: e19427
Received 2024 Nov 21; Accepted 2025 Apr 14
Copyright year: 2025
License: This is an open access article, free of all copyright, made available under the Creative Commons Public Domain Dedication. This work may be freely reproduced, distributed, transmitted, modified, built upon, or otherwise used by anyone for any lawful purpose.
License URL: https://creativecommons.org/publicdomain/zero/1.0/

Keywords: Pelvic floor, Ultrasound, Reliability of results, Nulliparity, Young adults, Female

Funding: The authors received no funding for this work.

==============================
Background

The levator ani muscle (LAM) plays a crucial role in maintaining pelvic floor function, providing support to pelvic organs and assisting in physiological processes such as micturition and defecation. Ultrasound imaging, particularly M-mode ultrasound, has been increasingly employed in the evaluation of pelvic floor muscle activity.

Objectives

The goal of the present study was to assess the intra- and inter-rater reliability of M-mode ultrasound measurements of LAM thickness, muscle activation speed, and contraction time during a 5-s sustained pelvic floor contraction in asymptomatic women.

Methods

Thirty-four healthy nulliparous women aged 18 to 25 were recruited for this cross-sectional study. Transabdominal M-mode ultrasound was used to measure the intra- and inter-rater reliability of LAM thickness during maximum pelvic floor contraction. Each participant performed three pelvic floor contractions, with measurements repeated twice.

Results

Intra-rater reliability ranged from poor to good reliability for LAM thickness (ICC = 0.621–0.899), with similar results for pelvic rise time and contraction velocity. Inter-rater reliability was excellent for LAM thickness (ICC = 0.910) but only fair for pelvic rise time and contraction velocity (ICC = 0.673–0.647).

Conclusions

M-mode ultrasound imaging demonstrates excellent inter-rater reliability for LAM thickness measurement and fair intra-rater reliability for other functional variables. This technique appears to be a promising method for evaluating pelvic floor muscle activity.

Introduction

The pelvic floor constitutes a critical anatomical element in the complex framework of the abdominopelvic system, acting as a functional unit that supports the abdominopelvic cavity (Prather, Spitznagle & Dugan, 2007). Composed of muscles, ligaments, and fasciae, this structure plays a central role in various physiological functions, from supporting pelvic organs to regulating urinary, defecatory, and sexual functions (DeLancey, 1993; Eickmeyer, 2017). In the anatomy of the pelvic floor, two main layers are distinguished: superficial and deep, each with specific and complementary functions. The deep layer includes key muscles such as the coccygeus and the levator ani muscle (LAM), the latter formed by the puborectalis, pubococcygeus, and iliococcygeus muscles (Bø, 2004). The precise interaction between these muscles and the connective tissues ensures adequate support of the pelvic organs and prevents dysfunctions (Lawson & Sacks, 2018). The integral function of the pelvic floor, together with the function of the diaphragm, abdominal and lumbar musculature, all components of the core complex, provide the stability required in the abdominopelvic compartment and favors the support of the pelvic organs, as well as their participation in essential functions such as micturition, defecation, and sexual activity (Huxel Bliven & Anderson, 2013). Specifically, the maintenance of this support function largely depends on the health and strength of theLAM, which act as a kind of sling to support the bladder, reproductive organs, and rectum (Muro & Akita, 2023). This functional complexity highlights the importance of understanding the detailed anatomy of these muscles to effectively address any dysfunctions that may arise (Huxel Bliven & Anderson, 2013). The biomechanical relationship between the pelvic floor musculature and the pelvic organs is crucial to understand their function. The interaction between the LAM and the connective tissues that attach the uterus and vagina to the lateral walls of the pelvis ensures normal support of the pelvic organs. This dynamic is essential to maintain the closure and elevation of the pelvic floor, preventing the descent of the pelvic organs and thus preventing dysfunctions (Muro & Akita, 2023). Weakness or injury in the pelvic floor musculature can manifest as pelvic organ prolapse, dysuria, defecatory dysfunction, and sexual dysfunction (Bø, 2004). These dysfunctions significantly affect the quality of life of individuals and highlight the need for effective preventive and therapeutic strategies (DeLancey, 1993). Ultrasound (US) emerges as a valuable tool in the functional evaluation of the pelvic floor musculature (Krasnopolsky, Ami & Dar, 2024). Different modalities, such as transabdominal and transperineal ultrasound, provide a detailed view of the muscle activity in real time (Zhu & Liu, 2025). M-mode ultrasound, specifically, offers a dynamic evaluation by capturing movement over time, allowing for the measurement of bladder displacement during maximum pelvic floor contraction (Arranz-Martín et al., 2021; Martínez-Bustelo et al., 2021; Sherburn et al., 2005; Sherburn & Bø, 2005; Zając et al., 2021).

This technique provides an objective and quantitative approach to assessing muscle activation speed, contraction time, and coordination, contributing to a more comprehensive understanding of pelvic floor function (Ide et al., 2022). Studies such as Nyhus et al. (2020) have demonstrated the reliability of ultrasound-based assessments of pelvic floor contractions, reinforcing the relevance of incorporating M-mode ultrasound for evaluating bladder base displacement as a functional measure of LAM activation (Nyhus et al., 2020). The application of US in clinical settings enhances our ability to detect dysfunctions and optimize rehabilitation strategies making it a promising tool in both research and practice (Ide et al., 2022). The B-mode format of transabdominal ultrasound as we have seen has been validated as a tool for measuring the displacement of the bladder base as an evaluation of the activation of the LAM (Zając et al., 2021). The incorporation of M-mode for the evaluation of this displacement would be key as an objective tool for such muscle activation.

The objectives of this study were to investigate the level of activity through a sustained contraction for 5 s of the pelvic floor musculature using M-mode ultrasound, measuring the change in bladder base displacement, the speed of activation, and the time of sustained contraction, and secondly to analyze the intra- and inter-examiner reliability of M-mode images of the activation of the pelvic floor muscle in healthy women.

Methods

Design

A reliability study was designed, according to the Strengthening the Reporting of Observational Studies in Epidemiology (STROBE) guidelines and checklist (Bossuyt et al., 2015).

Participants

The study included 34 healthy, nulliparous female participants, aged between 18 and 25 years (mean age: 21.9 ± 4.7 years). All participants were university students fluent in Spanish. Exclusion criteria comprised pregnancy, history of previous pregnancies, surgical procedures in the abdominal or pelvic regions, active pelvic floor disorders (such as urinary incontinence or pelvic pain), use of medications affecting muscle function (e.g., muscle relaxants, antidepressants), chronic systemic conditions (fibromyalgia, diabetes), or neurological disorders. Recruitment was conducted through posters at the university campus, inviting elegilble woman who meet the inclusion and exclusion criteria and requested to participate in the study were accepted.

Sociodemographic data are shown in Table 1.

Table 1 Sociodemographic data.

Data	Total = 67	
Age, years	21.90 ± 4.72	
Height (cm)	1.65 ± 0.05	
Weight (kg)	60.88 ± 8.84	
BMI	22.117 ± 2.75	
Note:

BMI, Body Mass Index. Data expressed as mean ± SD.

Ethical considerations

The present study followed the Declaration of Helsinki guidelines, and it was approved by the Ethics Committee of Universidad Europea de Madrid (CIPI/213006.52). All the participants read and signed the informed consent form prior to their participation.

Procedure

Participants were instructed to arrive with a moderately full bladder, achieved by avoiding urination for 1 h prior and consuming approximately ½ liter of water within half an hour before their appointment. They were positioned in a supine lying posture with a neutral pelvic alignment, and a small pillow supported the head while the lower limbs were semi-flexed, resting on a foam roller (Arranz-Martín et al., 2021; Martínez-Bustelo et al., 2022b).

The ultrasound examination utilized a convex C5-1 probe with a 5 MHz frequency (LOGIQe S7 Expert Ultrasound, GE Healthcare, Anaheim, CA, USA), set in M-mode. The evaluation was conducted by two physical therapist specializing in pelvic floor and abdominal ultrasound, with extensive clinical experience and a solid background in the acquisition and analysis of ultrasound imaging within quantitative research. The transducer was positioned transabdominally above the suprapubic region and angled approximately 15–30 degrees to the vertical axis to optimize visualization of the pelvic floor muscles and bladder base (Sherburn et al., 2005). Imaging was performed in transverse and sagittal planes, using anatomical markers such as the junction of the hyperechoic and hypoechoic lines, which delineate the deep pelvic floor structures (Martínez-Bustelo et al., 2022a).

Participants were instructed to perform a maximal pelvic floor muscle contraction by tightening the muscles “as if holding in gas” (Charlanes et al., 2021). To ensure familiarization, they practiced this action three times before data collection. During the assessment, they completed three maximal contractions, each sustained for 5 s, with 15-s rest intervals to prevent muscle fatigue (Fig. 1).

Figure 1 M-mode transabdominal ultrasound image of the bladder base during sustained pelvic floor activation.

Cranial displacement of the bladder base is observed upon activation.

First, the evaluation was conducted by Evaluator 1. Then, a 10-min rest period was allowed before repeating the procedure with Evaluator 2, ensuring interobserver reliability of the measurements.

Bladder displacement was evaluated by measuring the vertical movement of the bladder base from its resting position during maximal contraction. The distance in centimeters was measured between the initial resting position of the bladder base and the maximum position reached during contraction. This measurement reflects the pelvic floor elevation capacity and was determined as the average of the measurements obtained in the transverse planes (Laycock & Jerwood, 2001; Leitner et al., 2019).

Another recorded measurement was the velocity to reach the stable position of maximum bladder base elevation after the initiation of the contraction. This parameter allowed for the assessment of the neuromuscular efficiency of the LAM and the detection of possible delays in activation (Martínez-Bustelo et al., 2022a). Additionally, pelvic floor rise time was measured referring to the time it took for the pelvic floor to reach its maximum elevation during a voluntary contraction, providing insight into the activation speed of the pelvic floor muscles.

The time during which the bladder base remained elevated before initiating its descent was measured, reflecting the resistance capacity of the LAM to maintain muscular activation. Muscle fatigue was considered to occur when the bladder base began to descend due to loss of activation. This phenomenon was identified by a progressive reduction in the achieved height and, in some cases, a variation in the thickness of the ultrasound image, which could suggest a failure in the muscle’s supportive capacity (Martínez-Bustelo et al., 2022a). The measurement was performed three times for each participant (Fig. 2).

Figure 2 M-mode transabdominal ultrasound image of the bladder base during sustained pelvic floor activation, showing measurements of cranial displacement of the bladder base (d1), holding time (t1), and elevation velocity (v1).

Each measurement was taken twice in both transverse and sagittal views, and the average displacement was recorded.

Data analysis

An M-mode ultrasound image of the maximum contraction of the pelvic floor was obtained, and measurements of thickness were performed (Arranz-Martín et al., 2021) taking into account the point of maximum stable displacement. In the transverse plane, the lower line of the bladder was used as a reference (marked by the end of the anechoic margin and the beginning of the hyperechoic line representing the deep plane of the pelvic floor) (Arranz-Martín et al., 2021).

Statistics

The statistical analyses were conducted using SPSS software, version 22.0 (IBM Corp., Armonk, NY, USA). Results are reported as mean values alongside their standard deviations (mean ± SD). The Shapiro-Wilk test was utilized to evaluate the normality of the data distribution. The intraclass correlation coefficient (ICC), using a two-way mixed-effects model with an emphasis on consistency (alpha), was applied to assess both intra- and inter-rater reliability for the ultrasound measurements of LAM thickness, pelvic floor rise time, velocity, and contraction duration. A p-value of p < 0.05 was considered statistically significant, indicating that the ICC values were significantly different from zero. The classification for ICC values was as follows: values less than 0.50 indicate poor reliability, 0.50 to 0.75 indicate moderate reliability, 0.75 to 0.90 indicate good reliability, and values greater than 0.90 indicate excellent reliability (Koo & Li, 2016).

The standard error of measurement (SEM) was calculated using the formula: [SEM = pooled SD * (1−ICC)]. The minimal detectable change (MDC) was also computed for a 95% confidence level using the formula: [MDC = SEM * z * 2] using the standard normal z-score for a 95% confidence interval which is z = 1.96. The SEM represents the inherent error of the measurement instrument, indicating its precision, while the MDC defines the smallest detectable change in a score that can be interpreted as a true change beyond the measurement error threshold, with statistical significance set at p < 0.05 (Weir, 2005).

Results

Intra-rater reliability

The intra-rater reliability for the measurement of LAM thickness showed moderate reliability and good reliability between evaluators, 0.621 and 0.899, respectively. Moderate and good reliabilities were reported for both evaluators for pelvic rise time variable, 0.705 and 0.764. Regarding the velocity the reliability was 0.669 and 0.859 from moderate to good reliability. Contraction time reported good reliability with 0.815 and 0.762, respectively (Table 2).

Table 2 Intra- and inter-rater reliability between experienced rater 1 and experienced rater 2 ultrasonography measurements of the transversus abdominis during different testing conditions.

Measurement	Mean ± SD	ICC (95% CI)	SEM	MDC	
Intra-rater 1 reliability					
Thickness (mm)	0.809 ± 0.312	0.621 [0.335–0.796]	0.192	0.692	
Pelvic rise time (s)	0.391 ± 0.233	0.705 [0.482–0.841]	0.126	0.353	
Velocity (mm/s)	2.569 ± 1.273	0.669 [0.420–0.822]	0.733	2.588	
Contraction time (s)	2.090 ± 1.24	0.815 [0.675–0.900]	0.533	1.499	
Intra-rater 2 reliability					
Thickness (mm)	0.716 ± 0.256	0.889 [0.805–0.941]	0.085	0.304	
Pelvic rise time (s)	0.540 ± 0.321	0.764 [0.583–0.874]	0.155	0.437	
Velocity (mm/s)	1.766 ± 1.01	0.859 [0.750–0.925]	0.379	1.065	
Contraction time (s)	1.781 ± 1.35	0.762 [0.579–0.873]	0.658	1.837	
Inter-rater reliability					
Thickness (mm)	0.619 ± 0.370	0.910 [0.833–0.952]	0.111	0.311	
Pelvic rise time (s)	0.359 ± 0.265	0.673 [0.397–0.823]	0.151	0.421	
Velocity (mm/s)	1.431 ± 1.225	0.647 [0.348–0.809]	0.727	2.548	
Contraction time (s)	1.591 ± 1.326	0.673 [0.397– 0.823]	0.758	2.675	
Note:

mm, milimetres; Mm/s, milimetres per second; MCD, minimally detectable change; SEM, standard error measurement.

Inter-rater reliability

Thickness of the LAM was 0.910 with excellent reliability. The rest of variables reported moderate reliability (pelvic rise time = 0.673, velocity = 0.647 and contraction time = 0.673) (Table 2).

Discussion

The primary outcome of our study underscores the efficacy of M-mode ultrasound as a proficient method for assessing the thickness and functional characteristics of the LAM in asymptomatic individuals. The LAM is fundamental for the proper functioning of the pelvic floor, as it provides support and stability to the pelvic organs. Assessing this muscle is key in rehabilitation, especially for identifying potential dysfunctions that can lead to issues such as incontinence or pelvic organ prolapse (Ashton-Miller & DeLancey, 2007). Ultrasound has emerged as an essential tool in pelvic floor physiotherapy, enabling a precise and non-invasive assessment of muscle function in real time, facilitating both diagnosis and treatment monitoring (Whittaker et al., 2007). Its use is complemented by other clinical scales, such as the Modified Oxford Scale (MOS) (Newman & Laycock, 2008) and Laycock’s PERFECT scheme (Laycock & Jerwood, 2001), which evaluate the strength, endurance, and other contractile abilities of the pelvic floor. Dietz, Jarvis & Vancaillie (2002), explored the correlation between palpation (MOS) and perineometry compared to US, with highly significant correlations. In Dietz, Jarvis & Vancaillie (2002), the parameters observed were static (inclinations and angles), in contraposition with the M-mode performed in this study. This imaging technique’s reliability is also found in similar studies on other muscular complex, which demonstrated that M-mode features high intra- and inter-rater reliability when assessing muscle activity, regardless of the examiner’s level of experience (Dietz, Jarvis & Vancaillie, 2002; Romero-Morales et al., 2020). The use of M-mode ultrasound provides valuable data on the speed, thickness, and effectiveness of LAM contraction, offering a more comprehensive view of pelvic floor functionality and enabling more personalized and effective therapeutic interventions to improve functional recovery in patients at risk of pelvic floor dysfunctions (Laycock & Jerwood, 2001; Zając et al., 2021). Furthermore, the ability to dynamically track muscle integrity and function over time through non-invasive assessments has been explored by several authors, due to its value for the management of pelvic health (Gachon et al., 2021).

Despite their extended clinical use, both the MOS and the PERFECT scheme have shown some variability in results between evaluators, suggesting the need for more objective complementary tools (da Silva et al., 2021; Romero-Cullerés et al., 2019). In this regard, M-mode transabdominal ultrasound has proven to be a promising option, allowing for precise evaluation of bladder base displacement during levator ani contraction (Zając et al., 2021). This not only enhances the information obtained through digital palpation but also offers a useful biofeedback tool during the treatment of pelvic floor dysfunctions.

A notable finding in this study was that inter-rater reliability for LAM thickness was higher than any of the observed intra-rater reliability values. Inter-rater reliability measures agreement under identical conditions between two independent raters, both of whom followed a standardized protocol and conducted their assessments within a short time frame. In contrast, intra-rater reliability reflects measurements taken at different time points, where slight variations in participant movement, or muscle activation could introduce variability. This phenomenon has been observed in other studies, such as Cleary et al. (2022), Wallwork, Hides & Stanton (2007) and Pirri et al. (2019) where high inter-rater reliability was achieved for muscle ultrasound measures while intra-rater reliability values were slightly lower. This finding supports the observation that inter-rater reliability can sometimes surpass intra-rater reliability, potentially due to methodological factors such as standardized protocols followed by multiple raters and variations in individual assessments over time. The results of the present study show that LAM thickness measurement is highly reliable, with excellent inter-rater reliability and moderate-to-good intra-rater reliability. However, functional parameters such as pelvic rise time, velocity, and contraction time showed lower reproducibility, particularly in inter-rater assessments. Thus, future research should explore factors that influence measurement consistency, such as evaluators’ training, sustained probe positioning, and muscle fatigue standardization. This is especially critical in clinical environments where precise and reproducible measurements are vital for accurate diagnosis and management of pelvic floor disorders. These findings are corroborated by several (Krasnopolsky, Ami & Dar, 2024) studies. Rostaminia et al. (2015) using 3D endovaginal ultrasound, found also fair sensitivity (60%) and specificity (63%) for discriminating LAM deficiency. The present study was conducted in healthy individuals, therefore, the potential use of M-mode in samples such as Rostaminia et al. (2015) study should be explored.

However, our findings also highlight some variability in intra-rater reliability across different measures, suggesting that while ultrasound is a highly effective diagnostic tool, factors such as the examiner’s technique and the participant’s consistency in performing muscle contractions can influence measurement outcomes. These considerations are coincident with findings from other muscle complexes, such as the soleus, where muscle positioning significantly influenced outcomes, demonstrating the need for consistent protocol application (Romero-Morales et al., 2020).

The present study has several limitations that should be considered when interpreting the results. First, the major limitation of the study is design is the small number of participants. Another limitation is that the reliability of this method is dependent on the patient’s ability to contract their pelvic floor muscles efficiently and for at least 5 s. Moreover, the study’s focus on asymptomatic individuals limits the generalizability of the findings to populations with existing pelvic floor disorders or symptoms. Future research should include diverse participant groups to validate the utility of M-mode ultrasound across different clinical scenarios. Finally, our study did not explore the effects of different positioning of the pelvic floor muscles during ultrasound assessment, which could potentially affect muscle activity and measurement outcomes. Investigating these positional influences could provide deeper insights into the optimal protocols for ultrasound assessments of the LAM.

Conclusions

The results of the present study suggest that measuring the LAM thickness with ultrasound imaging have moderate to good intra-rater reliability as well as excellent inter-rater reliability. Considering the M-mode variables, intra- and inter-rater reliability show moderate to good reliability.

Supplemental Information

Supplemental Information 1 Raw Data.

Supplemental Information 2 STROBE checklist.

Supplemental Information 3 CONSORT checklist.

Additional Information and Declarations

Competing Interests

The authors declare that they have no competing interests.

Author Contributions

Esther Delgado-Pérez conceived and designed the experiments, performed the experiments, authored or reviewed drafts of the article, and approved the final draft.

Alberto Bermejo-Franco performed the experiments, authored or reviewed drafts of the article, and approved the final draft.

Raquel Díaz-Meco conceived and designed the experiments, authored or reviewed drafts of the article, and approved the final draft.

Vanesa Abuín-Porras analyzed the data, prepared figures and/or tables, and approved the final draft.

Carlos Romero-Morales analyzed the data, prepared figures and/or tables, and approved the final draft.

Laura González-Fernández conceived and designed the experiments, authored or reviewed drafts of the article, and approved the final draft.

Mónica De-La Cueva-Reguera conceived and designed the experiments, performed the experiments, authored or reviewed drafts of the article, and approved the final draft.

Human Ethics

The following information was supplied relating to ethical approvals (i.e., approving body and any reference numbers):

The Ethics Committee of European University granted ethical approval to carry out the study (Ethical Application Ref: CIPI/213006.52) in accordance with the Declaration of Helsinki guidelines.

Data Availability

The following information was supplied regarding data availability:

The raw measurements are available in the Supplemental File.

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
