# Peer review of "Intra- and inter-rater reliability of ultrasound measurements of the levator ani muscle thickness in and M-mode in asymptomatic individuals: a cross-sectional study"

_PeerJ, doi:10.7717/peerj.19427_

## Round 0.1 · original submission · Major Revisions

· Academic Editor

Major Revisions

Dear Authors,
As our reviewers mentioned many points to be addressed in the present form of the manuscript, I invite you to make a major revision of your paper. Please do the needful and resubmit asap. All the best.

Reviewer 1 ·

Basic reporting

This study is a intra and inter-rater reliability study assessing levator ani muscle thickness, activation speed and contraction via ultrasound. The manuscript is clear and easy to read. Literature references are provided. However there is no figure showing how to image levator ani muscle which is a really important evidence to understand how well the study conducted. Another drawback is lack of tables that summarize the measurements (Muscle thickness and the others) and ICC values.

Experimental design

The major limitation of the study is design is small number of participants which should be declared as a limitation.

Validity of the findings

The classification for ICC values should be revised according to literature (1).

1. Koo TK, Li MY. A Guideline of Selecting and Reporting Intraclass Correlation Coefficients for Reliability Research. J Chiropr Med. 2016 Jun;15(2):155-63. doi: 10.1016/j.jcm.2016.02.012. Epub 2016 Mar 31. Erratum in: J Chiropr Med. 2017 Dec;16(4):346. doi: 10.1016/j.jcm.2017.10.001. PMID: 27330520; PMCID: PMC4913118.

Additional comments

No additional comments.

·

Basic reporting

The manuscript has a good level of English, however, there are several grammatical mistakes that need to be addressed.
For example: line 82 (was design -> was designed), line 78 (.. -> .), line 137 brackets are not complete.

A key aspect in the methodology is repeated (almost perfectly) in the text, (Methods: Lines 109: 121 and lines 123:131). Authors need to re-read the manuscript to ensure no repetition further and adjust the text.

Several key references and support are missing from several statements in the Introduction (including lines 62-64, ultrasound as a valuable tool in the functional evaluation of the pelvic floor, authors only reference the measurement of bladder displacement later in line 66). Authors also do not provide support for the incorporation of M-mode for the evaluation of bladder base displacement, this could be a similar structure or assessment of another anatomical region.

Authors should also reference: Nyhus, M.Ø., Oversand, S.H., Salvesen, Ø., Salvesen, K.Å., Mathew, S. and Volløyhaug, I. (2020), Ultrasound assessment of pelvic floor muscle contraction: reliability and development of an ultrasound-based contraction scale. Ultrasound Obstet Gynecol, 55: 125-131. https://doi.org/10.1002/uog.20382.

As Nyhus et al. determine intra-interrater reliability and agreement of ultrasound measurements of pelvic floor contraction (focusing on the levator hiatal anteroposterior diameter and hiatal area. This will help the authors understand how measurement on M-mode compares.

Authors do not include any figures in the text; however, I do think the paper needs a figure to demonstrate the process for measuring the levator ani muscle thickness. Authors currently refer to reference 9, however, for reproducibility and transparency authors should either state the protocol from reference 9 or describe and visualise (with a diagram) the process here.

Experimental design

The authors set out to assess several objectives being to investigate the level of activity through a sustained contraction, and to analyse inter and intra examiner reliability of M-mode images of thickness of the pelvic floor muscle in healthy women. This is suitable to PeerJ. The research question is reasonably well-defined, but the authors lack connecting the research question to clinical relevance and a suitable clinical application of pelvic floor muscle thickness measurement. The authors describe that they developed the study in accordance with STROBE guidelines, which should help to standardise the work.

The ultrasound acquisition is adequately described, and readers are referred to reference 15, again if this is a relatively new measurement and/or assessment a figure of the experimental setup would be beneficial in the section (Procedure).

Unfortunately, authors do not provide sufficient detail to recreate the experiment, as mentioned above a visualisation and specific protocol for muscle measurement should be referenced explicitly or described here (ideally as a figure). There is little explanation of the point of maximum stable displacement.

The authors also do not explain how they measured the time of sustained contraction, change in thickness or speed of activation. These are key objectives mentioned in 106-110 of the introductions, but there is no mention on the acquisition of this data in this manuscript.

The authors also do not provide any information of the evaluators (e.g., level of experience, years of using M-mode imaging, training required to perform this analysis, time taken to perform the analysis). This lack of detail makes it difficult to reproduce the study with confidence.

Furthermore, in the methods, authors do not describe the inter or intra -evaluated studies, they do not mention the number of evaluators, which evaluator performed the intra-rater reliability study or if all did. There is no description on whether the evaluators were blinded to previous analysis, or each other’s assessment which is required to minimise bias. The time interval between measurements for intra-rater reliability is also not specified. An unbiased study should include a reasonable gap (e.g., weeks) to avoid recall bias.

Regarding the statistics section, the formulas should be properly formatted, and the standard normal z-score for 95% confidence noted as z should be specified (e.g., 1.96). The equation for SEM should insure that (1-ICC) is all within the square root function.

It is unclear if the minimal detectable change is an interval or absolute value.

The classification for ICC values lacks citation, and rationale for not referring to a widely accepted standard should be given, that is: Koo TK, Li MY. A Guideline of Selecting and Reporting Intraclass Correlation Coefficients for Reliability Research. J Chiropr Med. 2016 Jun;15(2):155-63. doi: 10.1016/j.jcm.2016.02.012. Epub 2016 Mar 31. Erratum in: J Chiropr Med. 2017 Dec;16(4):346. doi: 10.1016/j.jcm.2017.10.001. PMID: 27330520; PMCID: PMC4913118.

The authors note a p-value but not describe the relevance or a statistical test or hypothesis, which is counterintuitive. The authors also do not mention whether multiple testing correction is used, this reduces the risk of Type 1 error and should be applied if multiple reliability tests are being conducted for example the ICC for different parameters like pelvic floor muscle thickness, speed of activation if they are all inferred from the M-mode ultrasound.

Also to note, in the methods section authors seem to exchange the observable names between pelvic floor muscle thickness to LAM muscle thickness, speed of activation to velocity, time of sustained contraction to contraction duration. This must be consistent throughout the text. In addition, a new parameter called pelvic floor rise time is used and this is not explained or introduced during the text. This lack of details makes it difficult to reproduce the study.

Validity of the findings

The authors report on the inter and intra rater measurements briefly within the results section, confidence intervals of the ICC are missing. A concern is that the authors do not report on the level of activity, speed of activation, time of sustained contraction or pelvic floor rise time in the results. They only report on the inter or intra rater reliability of these measurements.

In addition, despite the statistics section in Methods, the authors do not report on the minimal detectable change, or the standard error of measurement, or the significance.

A key concern is that the intra-rater reliability (0.621, 0.899) is substantially lower than the measured inter-rater reliability (0.910) which is very counterintuitive. I would suggest there is a mistake in the calculation of the ICC/raw data or assume one evaluator was provided with feedback by the other, meaning their second evaluation differed sustainably from the first but then that the agreement between evaluators improved. This is why the experiment details of the intra and inter evaluator studies are necessary in the Methods.

Within the discussion, authors note that M-mode is proficient for assessing the thickness and functional characteristic of the LAM, however, this is not explicitly reported in their results section. The results only measure the inter-intra evaluator ICC scores. The discussion then goes into an in-depth literature review rather than focusing on the key contributions/results of this paper, this should be restructured to focus on the study's contributions.

In line 223, the authors note that the ICC values were consistently above 0.90, however, only the inter-rater score was over this limit, and as mentioned above this result seems counterintuitive and should be checked by the authors or explained as counterintuitive in the text. The claim that ICC values are consistently above 0.90 is therefore, misleading to the reader.

Authors correctly identify the limitations of the study, being that the population is limited to asymptomatic individuals. The link could also be made that the sample size of 34 may be limited, and that the reliability of this method is dependent on the patient’s ability to contract their pelvic floor muscles efficiently and for at least 5 seconds.

Additional comments

This study is indeed needed to understand the reproducibility of ultrasound measurements of levator ani muscle/pelvic floor muscle thickness. It has several strengths such as being clearly written. It also is an interesting study and one that is important to share with the community.

However, there are some key concerns within the text regarding missing references, detail lacking in the methods section (e.g., inter intra study set up, measurement of LAM thickness and observables, statistical design choices), observables not reported in the results and counterintuitive results where inter-rater reliability fairs much higher than intra-rater reliability, without any investigation in the discussion section or without the raw data displayed in this text as a table (i.e., average LAM thickness or difference in LAM thickness between observers).

This makes it difficult to recreate and to apply these results and set up with existing or new studies.



A summary of all key revisions needed:

Grammar and language: correct grammatical mistakes, fix punctuation issues, address incomplete brackets.

Repetition in Methodology: avoid repetition of key aspects, review manuscript to ensure no further redundancy.

References and citations: add missing references, particularly in the introduction (ultrasound in pelvic floor evaluation). Include the citation by Nyhus et al. regarding ultrasound assessment of pelvic floor muscle contraction.

Figures: Include a figure/diagram to demonstrate the process for measuring levator ani muscle thickness for reproducibility.

Details of Experimental set up: Provide more detail on inter-intra evaluator reliability studies (e.g., number of evaluators, experience, blinding, time interval between measurements, level of training before analysis). Explain how the time of sustained contraction, change in thickness, and speed of activation were measured.

Clarification of terminology: ensure consistency in terminology (e.g., LAM thickness, speed of activation, contraction duration). Explain all terms (pelvic rise time).

Statistical Analysis:
Properly format formulas, ensure the equation for SEM is formatted correctly. Clarify whether minimal detectable change is an interval or absolute value. Explain rationale for why ICC classification values differ from standards (Koo & Li, 2016). Provide details on p-values, statistical tests and hypothesis used. Address if multiple testing corrections were applied to reduce Type 1 error.
Inconsistence in results and discussion:
Investigate the counterintuitive finding of inter-rater reliability being higher than any observed intra-rater reliability and explain in the discussion.
Report on confidence intervals for ICC scores and ensure all observables are reported in the results if they are an objective in the Introduction of the study.
Restructure the discussion to focus on the paper’s findings, rather than an extensive literature review.

Raw data: Present raw data in tables (e.g., LAM thickness, difference between observers and others) for transparency.

Study limitations: Discuss limited sample size and acknowledge the potential impact of this on the study’s generalisability as well as drawing the link to clinical impact.

---

## Round 0.2 · Minor Revisions

· Academic Editor

Minor Revisions

Dear authors, our reviewers suggested few more points to be addressed. Please do the needful and resubmit asap.
All the best

Staff Note: The figure titles and legends should be entered in the article metadata.

Reviewer 1 ·

Basic reporting

The revisions are improved the manuscript and the paper can be published with a minor addition. I couldn't see Figure legends, if the authors adds it to manuscript the paper will be complete.

Experimental design

No comment

Validity of the findings

No comment

Additional comments

No comment

---

## Round 0.3 · accepted · Accept

· Academic Editor

Accept

Dear authors,

With pleasure, I inform you that our reviewers have appreciated your efforts and opined to accept the manuscript in its current form.

Please remember it is just an academic acceptance and needs some editorial tasks to be completed before its publication. So, I advise you to be available for a few days to avoid any delays.

I wish you all the best for your future submissions in PeerJ.

Good luck

Reviewer 1 ·

Basic reporting

Figure legends are provided. No further change is needed.

Experimental design

Figure legends are provided. No further change is needed.

Validity of the findings

Figure legends are provided. No further change is needed.

Additional comments

Figure legends are provided. No further change is needed.